# Identifying Suitable Variables for Visual Aesthetic Quality Assessment of Permanent Forest Reserves in the Klang Valley Urban Area, Malaysia

**Riyadh Mundher** [1],*, **Shamsul Abu Bakar** [1], **Suhardi Maulan** [1], **Hangyu Gao** [1], **Mohd Johari Mohd Yusof** [1], **Azlizam Aziz** [2] **and Ammar Al-Sharaa** [3]

[1]  Department of Landscape Architecture, Faculty of Design and Architecture, Universiti Putra Malaysia, Serdang 43400, Malaysia; shamsul_ab@upm.edu.my (S.A.B.); suhardi@upm.edu.my (S.M.); gaohangyu1989@gmail.com (H.G.); m_johari@upm.edu.my (M.J.M.Y.)
[2]  Department of Recreation and Ecotourism, Faculty of Forestry and Environment, Universiti Putra Malaysia, Serdang 43400, Malaysia; azlizam@upm.edu.my
[3]  Department of Architecture, Faculty of Built Environment, University of Malaya, Kuala Lumpur 50603, Malaysia; ammoratawama@gmail.com
*  Correspondence: arch.riyad@gmail.com

**Abstract:** Permanent forest reserves (PFR) in urban areas are an integral aspect of the urban forest concept and the basis of a city's green infrastructure. The preservation of natural forests in urban areas has become a widely researched topic due to the environmental, social, and economic benefits provided by it. Although several studies have shown that visual aesthetics play a role in preserving natural forests in urban areas, visual aesthetic value is typically not prioritized in preservation plans since it varies based on the physical characteristics of natural forests in urban areas, making it difficult to measure universally. Therefore, this research aims to identify the suitable variables for assessing the visual aesthetic quality of permanent forest reserves within urban areas in Malaysia. This study selected two permanent forest reserves based on four criteria. Data were collected via participant-generated images taken along selected forest trails based on participant visual aesthetic preferences. Researchers and experts analyzed and classified the data according to content to identify suitable visual aesthetic quality variables. This research identified 14 suitable variables for assessing the visual aesthetics of PFRs in urban areas, with a dominant preference for information-processing theory variables. Mystery was the most present variable for the visual aesthetic quality assessment of PFRs. Furthermore, participants' educational and emotional backgrounds, categorized as design, environmental, social, and technical, impacted their visual aesthetic preferences. Our findings serve as a foundation for assessing the visual aesthetic quality of natural forests within urban areas in Malaysia.

**Keywords:** urban forests; urban area; information-processing theory; coherence; complexity; legibility; mystery; openness; uniqueness; participant-generated image

## 1. Introduction

Urban green space is a vital component of city infrastructure because it provides benefits and services to urban residents, such as climate change mitigation and adaptation, biodiversity conservation, and improvements to human health and well-being [1–3]. Specifically, natural forests in urban areas enhance urban environmental quality by purifying the air, reducing pollution, creating shade, retaining rainfall, conserving energy, providing habitat for biodiversity and local wildlife, and providing aesthetic enjoyment and recreational opportunities [3–5]. Natural forests in urban areas contribute to local economic development, increase property prices, and attract economic and recreational activities. For example, several studies have shown that apartment prices are significantly

higher in the presence of natural forests in urban areas [3,6,7]. Natural forests in urban areas benefit the urban population's health and well-being, by removing the stress of noise associated with urban life, particularly as urban life is characterized by the constant stress of excessive concrete, flashing signs, and bright colors. Visiting natural forests in urban areas and enjoying their scenic beauty can re-energize urban residents and alleviate tension and stress [3,6,8,9].

A growing number of international organizations and global sustainability policies recognize the importance of natural forests in urban areas. This is highlighted by the United Nations General Assembly's Sustainable Development Goal 11, which calls for "making cities and human settlements inclusive, safe, resilient, and sustainable." Similarly, the First World Forum on Forests presented a vision of how cities across the world could use forests to make cities greener, healthier, and happier places to live [1]. Despite international recognition of their benefits, natural forests in rapidly developing urban areas face a constant threat of conversion into residential or commercial property to suit the needs of the growing urban population [2]. According to a United Nations estimate, the percentage of the world's population living in urban areas will grow to 66% by 2050, indicating that continued urbanization will have a negative impact on urban natural forest areas [10]. Therefore, many countries, such as Malaysia, have enacted laws and policies to protect forests. The Malaysian Government established a National Forestry Act, under which the government has the power to establish, protect, and maintain any forest through the Official Gazette. Under this law, the Department of Forestry of Peninsular Malaysia (FDPM) classifies these forests as permanent forest reserves (PFRs) [11].

Permanent forest reserve (PFR) is a legislative designation created by the FDPM in accordance with the National Forestry Act of 1984. Its aim is to preserve and manage the natural forests by designating them as forests reserve. Permanent forests reserves are administered and classified the natural forests as production forest, protection forest, research and education forest, and amenity forest [12]. The current legislation focuses on ecological, economic, social, and educational aspects, while other aspects of forest preservation, such as visual aesthetics, remain vague and are not fully emphasized in the Forest Protection Act, despite being recognized as a strong motivator for protecting natural areas [3]. Mundher [12] investigated the values of visual aesthetics to promote the preservation of the Malaysian Permanent Forest Reserve within the existing legal framework. This study found that visual aesthetics are considered a crucial aspect of forest classification and preservation. Additionally, numerous previous studies have confirmed that visual aesthetic value plays an important role in the preservation and protection of forest areas considered exceptionally beautiful [13–16]; therefore, it should be included in the Forest Protection Act. However, physical variations between forests make it challenging to create a global unified framework for assessing forest visual aesthetics based on standardized variables. Therefore, this research aims to identify suitable variables for the visual aesthetic quality assessment of PFRs in urban areas based on local landscapes' character and local perception in Malaysia.

### 1.1. Natural Forests in Urban Areas

Natural forests in urban areas are part of urban forests close to cities, towns, and sub-urbs where people reside and work. Their presence within or near human population-dense city centers distinguishes them from natural forests within non-urban environments [6,17–19]. The definition of natural forests in an urban area is based on population density; however, there is no official minimum standard for population density near forests to be considered an urban forest or natural forest in an urban area. The urban residents typically use the term "natural forest" more commonly than "urban forest" even though they have more experience with urban forests than natural forests outside cities. In Malaysia, the concept of the term "urban forest" is not new, but the focus has traditionally been on natural forests and forest plantations; however, the use of this term in research has been steadily increasing in recent times, indicating a growing interest in this field [20]. The

rise in urban forest research can be attributed to accelerated urbanization and a growing awareness of the need for sustainable development in Malaysia. In general, areas of natural forests in urban settings can be referred to by various terms, such as urban forest natural areas, forest patches, urban woodlands, or urban forested parks [18].

Despite their similarities, natural forests in urban areas have several features that differentiate them from natural forests. The natural forests within an urban setting offer a similar image in terms of trees, shrubs, and an upper canopy, but are usually less dense than a natural forest. Natural forests in urban area assemblies are frequently regarded as new and complex because they feature a mix of human-cultivated, naturally regenerating, and non-native plant species [19,21]. Human intervention, such as pruning trees, is a characteristic that identifies natural forests in urban areas. Forests must be managed to provide a safe environment, enhance aesthetic value, and prevent interference with neighboring infrastructure, such as buildings, alleys, walls, signs, and light poles [22]. Natural forests in urban areas may contain paved trails and buildings that are not seen in natural forests. Occasionally, high-rise buildings can shade natural forests in urban areas, limiting the size, shape, and lifespan of trees in certain areas. However, trees in high-quality natural forests in urban areas may have unrestricted growth and maintenance by municipal forestry departments in the form of regular pruning, insect control, and fertilization. Consequently, high-quality tree growth environments would result in the growth of long-lived trees, which would bring substantial benefits to an urban area [18].

The characteristics differentiating natural forests in urban areas from natural forests can be summarized as diversity, connectedness, and dynamics [23]. Diversity is an outstanding characteristic of urban forests. Urban forests typically exhibit diversity for several reasons, all of which are connected to decisions and goals in urban forestry management. Human preferences and socioeconomic conditions play a significant role in shaping management choices concerning urban forest diversity. The diversity of urban forests can be observed across different land-cover types, including green areas, bodies of water, and man-made structures. The decision to plant various species contributes to the plant diversity found in urban forests, resulting in a blend of natural and cultivated plant species. Urban forests also incorporate a variety of human-made structures such as paved trails, signage, and rest areas. Additionally, there are multiple water elements present, such as rivers, lakes, streams, canals, and man-made ponds. Ultimately, urban forests are typically diverse due to a wide range of goals and complex management decisions, setting them apart from the diversity found in natural forests [18,21,23,24]. Connectedness in the urban environment is considered another key attribute of urban forests, as it describes the intensity and consistency between different components. This includes the connections between forests and other urban elements such as roads, buildings, and parks. Integrating urban forests with urban infrastructure, such as planting and maintaining trees within urban forested areas and paving sidewalks and roads, can enhance connectedness. Additionally, the connectedness between urban forests and other landscapes has become a crucial characteristic in mitigating the negative consequences of urban forest fragmentation, thus, positioning forests as a significant aspect of urban design and planning [10,23,25]. Urban forest dynamics refers to the transformations that occur in vegetation and other components over time. The strength and speed of urban forest dynamics are more closely linked to human activity rather than tree growth, setting it apart from natural forests. For instance, tall buildings can cast shade on urban forests, limiting the size, shape, and lifespan of trees in certain areas. Conversely, trees in urban forests benefit from high-quality growth resulting from regular maintenance practices such as pruning, insect control, and fertilization implemented by urban forest management. Consequently, the intricate coupling of urban forest dynamics with the rapid forces of human-induced change makes the management of urban forests complex and challenging [18,23,26].

*1.2. Variables of Visual Aesthetic Quality Assessment Captured from Theories*

For many centuries, aesthetics has been related to the philosophy of art based on the perception of aesthetics through artwork only [27]. However, since the 1960s, various theories and approaches have been developed in the field of natural visual aesthetics, many of which are based on physical and psychological natural features [3,28,29]. One of the most commonly studied theories in natural visual aesthetics is information-processing theory, which is based on four "informational variables" and their relationships: coherence, complexity, legibility, and mystery [30,31]. The theoretical model is founded on the idea that in natural circumstances, humans have two primary needs: understanding and exploration. These needs apply to what is currently perceptible (two-dimensional plane) or to what would be perceived if someone relocated to a different location (three-dimensional space). As a result, the matrix model is built on the intersection of two basic needs (understanding and exploration) and two levels (immediate and promised). The intersection used for informational variables is coherence (immediate understanding), complexity (immediate exploration), legibility (promised understanding), and mystery (promised exploration) [32–34]. For natural forests in urban areas, all four information variables can serve as visual aesthetic quality variables [33,34].

The prospect–refuge theory explains the concept of a victim's ability to see potential threats without being seen by them [35]. In forest spaces, the prospect is an environment that allows for an unobstructed view and is defined by natural openness. The prospect degree is described as the depth and openness of view, suggesting that the users will be drawn to large vistas [36,37]. Users prefer prospects, which can be direct or indirect. A direct prospect indicates that there is a vast field of vision that can be viewed directly from the chosen observation point. Panoramas, vistas, and peepholes are forms of direct prospects. A panorama is a view that can be observed from 180°, even if some minor impediments are present. A vista denotes a perspective line of sight with borders on both sides that resemble a frame. A peephole is an opening overlooking broad scenery, such as scenery in the vertical and horizontal borders overlooking a valley. An indirect prospect is when the user notices another point that indicates an extension of the field of view. The scenes forming an indirect prospect are the same as in a direct prospect, but with the vision skewed [38]. Openness can serve as a visual aesthetic variable for natural forests in urban areas, as it likely influences perception and preference for the visual aesthetic quality [33,34].

Dann [39] established the push–pull theory as the motivation forces that influence the perception of a place as visually attractive. In the theory, the push–pull factors are the starting points of an individual's decision process and important constructs for understanding human behavior. "Push" factors are the psychological drivers of behavior, while "pull" factors are external, cognitive incentives such as the attractiveness of a destination's unique feature. Although these factors have been utilized in research for many years, they have typically been explored in the context of tourism. In tourism, "push factors focus on whether to go, whereas pull factors focus on where to go" [40]. By focusing on the attraction of a destination's unique features, the pull concept mirrors the concept of uniqueness. Uniqueness implies that aesthetic assessment depends on whether a destination has uniquely identifiable features [33,34]. Destinations with unique traits are viewed as beautiful and have a pull force [41].

Aesthetic care theory refers to forest components that are immediately identifiable as created, signaling continued human presence in caring for the natural forests in urban areas [42]. Ecologically sound forests that elicit feelings of delight and acceptance are more likely to be perpetuated over time with sufficient human care, emphasizing the importance of natural visual aesthetic care in attaining cultural sustainability [43,44]. This theory is a cultural preference theory, which is consistent with the idea that natural visual aesthetics are subjective and dependent on an individual's culture. The aesthetics care theory is based on the cultural effect of human perceptions, ideas, attitudes, and behaviors, as well as the possibility of a local citizen becoming a forest aesthetic care sponsor. Pruning, shearing,

and removing foreign plants are frequently included in care procedures. The term "clean and green" has recently become popular within the idea of care since clean natural forests in urban areas exemplify care's aesthetics [45]. As a result, cleanliness can serve as a visual aesthetic variable in natural forests in urban areas since it may be suitable for studying cultural preferences toward visual aesthetics [33].

In summary, we identified seven primary variables of visual aesthetics via aesthetic theories: coherence, complexity, legibility, mystery, openness, uniqueness, and cleanliness (Table 1). These variables from the literature were used and explain as a foundation for variables linked with the aesthetic visual qualities of urban forests [33]. These variables will be evaluated to determine those that are suitable for assessing the visual aesthetic quality of permanent forest reserves in the Klang Valley urban area, Malaysia.

**Table 1.** The integrated theoretical framework of visual aesthetic variables [3,33,34].

| No. | Variable | Description | Theory |
|---|---|---|---|
| 1 | Coherence | The scene's elements are unified to produce a sense of order and cohesion. | Information-processing theory [29,30] |
| 2 | Complexity | The scene's elements are diversity or richness non-chaotic to provide a strong sense of variety. | Information-processing theory [29,30] |
| 3 | Legibility | The scene has a strong feeling of direction and ease of accessibility. | Information-processing theory [29,30] |
| 4 | Mystery | The scene has a motivation and thrills the viewer to discover hidden elements. | Information-processing theory [29,30] |
| 5 | Openness | The scene has a wide or panoramic view from which an observer can feel the vastness of the area. | Prospect–refuge theory [33] |
| 6 | Uniqueness | The scene's elements provide a distinct visual impression and are memorable. | Push–pull theory [37] |
| 7 | Cleanliness | The scene exhibits the aesthetic care and cleanliness that increases its quality without exceeding normal limits. | Aesthetics care theory [42,43] |

## 2. Materials and Methods

### 2.1. Study Areas

Klang Valley consists of five districts, beginning with Kuala Lumpur and extending to the four neighboring districts in the state of Selangor (Hulu Langat, Gombak, Petaling, and Klang). The valley has a total area of 2832 km2 and a population of over 8.2 million as of 2021, accounting for 25% of the Malaysian population. Its population is projected to exceed 10.4 million by 2035. As peninsular Malaysia's most urbanized region, Klang Valley serves as the country's industrial and commercial hub, contributing approximately one-third of the country's gross domestic product. The urbanized area is defined as the territory within 15 km of the locational center of gravity (LCG) in the Klang Valley, which includes both Kuala Lumpur and Petaling. The LCG is a region with a high level of human activity and accessibility, as well as a high rate of return on real estate investments, resulting in a high urban density [46]. As a result of this urban development, there are only some forests left, with a few small, scattered forests that have been protected as PFR within the territory of the LCG area in Klang Valley (Figure 1).

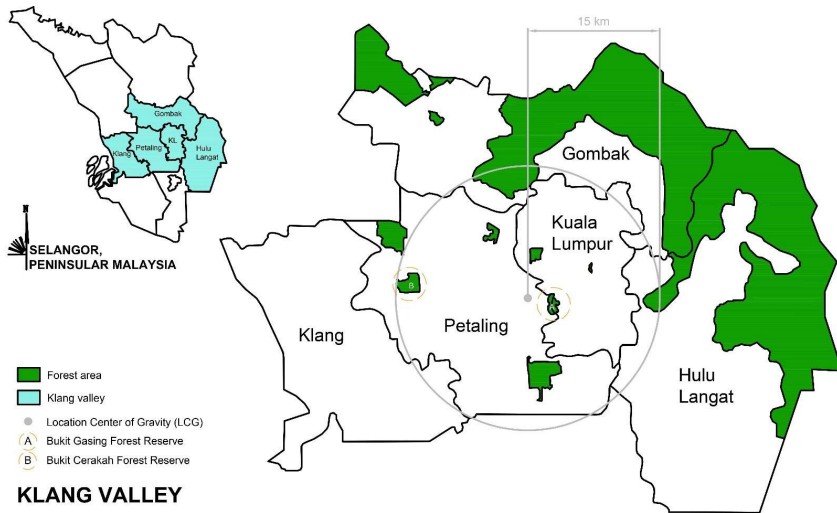

**Figure 1.** The location of Klang Valley and the study locations.

We utilized four criteria to select the study areas: (1) the forest must be in a densely urbanized area within 15 km of the LCG in Klang Valley; (2) the forest must be classified as a PFR; (3) the forest has man-made, paved trails; and (4) the forest is open and accessible to the public. The Bukit Kiara Forest and Bukit Persekutuan Forest were omitted since they were not PFRs. The Sungai Besi Forest Reserve, Bukit Lagong Forest Reserve, Hutan Simpan Ayer Hitam, and Bukit Sungai Puteh Forest Reserve were omitted due to a lack of man-made paved trails for assessment in the forest. The Bukit Nanas Eco Forest Reserve was omitted due to its closure for restoration work. Ultimately, we chose the Bukit Gasing Forest Reserve and Bukit Cerakah Forest Reserve for this study since they met all four criteria (Figure 2). We analyzed the selected forests to see how well they matched the characteristics of natural forests in urban areas.

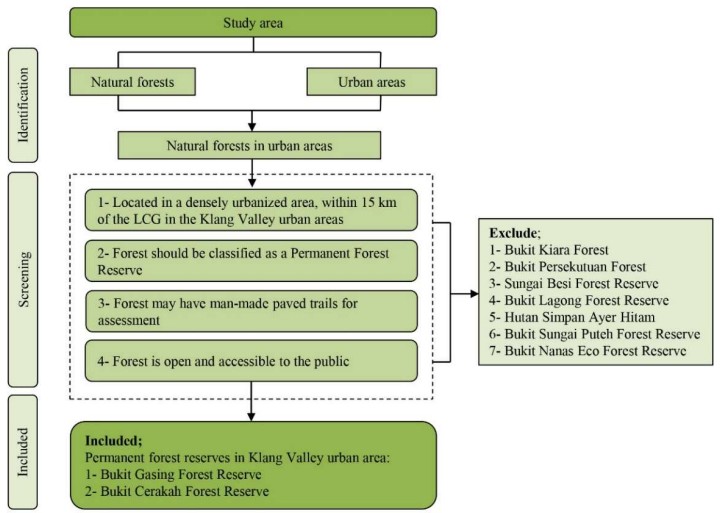

**Figure 2.** A flow diagram summarizing how the study areas were selected.

In this study, Bukit Gasing Forest Reserve and Bukit Cerakah Forest Reserve were selected as the study areas. The selected forests were described to see how well they matched the characteristics of natural forests in urban areas (Table 2):

**Table 2.** A description of the selected study area.

| Study Area | Description | Map |
| --- | --- | --- |
| (A) Bukit Gasing Forest Reserve | Bukit Gasing Forest Reserve is a 153-hectare tropical forest reserve located between the Petaling District (Selangor state) and Kuala Lumpur boundaries (federal). Consequently, the reserve is administered by two distinct governments: the Kuala Lumpur City Hall and the government of Selangor. A portion of the forest in Kuala Lumpur is connected to the Taman Rimba Bukit Kerinchi Park, which has led to the diversification of plant species. Bukit Gasing is classified as a PFR and a functional research and education forest. However, despite the aspirations of those responsible for proclaiming the forest a reserve, just a small area of the forest in Kuala Lumpur has been safeguarded as a PFR. The remainder of the forest was turned into residential and commercial developments, and urban threats continue to decrease the forest's size. | 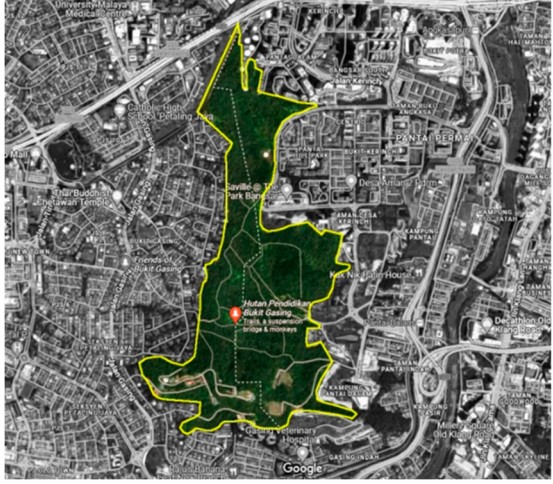 |
| (B) Bukit Cerakah Forest Reserve | Bukit Cerakah Forest Reserve is an approximately 800-hectare tropical forest reserve located in Petaling District (Selangor state). Formerly encompassing several thousand hectares, it has already lost a significant amount of land, as much of the low-lying forest was converted into housing areas. A solitary remnant of the original rainforest is concentrated primarily on the hills, while the lowlands are also home to botanic gardens, resulting in a diversity of plant species and a connectedness between the forest and botanical gardens. Bukit Cerakah is classified as a PFR and a functional protection forest. | 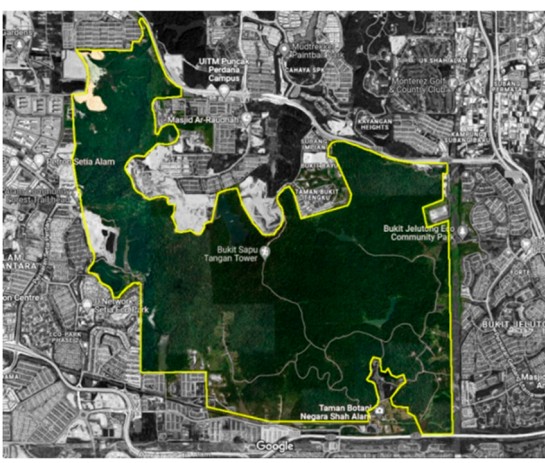 |

## 2.2. Data Collection

The experiment was carried out in December 2021, when the weather was pleasant and there was no rain; five hours each day in two days for each forest. Initially, the plan was to seek forest visitors to help, but just one agreed, and the results were disappointing. Therefore, the researchers posted a notice on forest community Malaysia's social media pages to solicit volunteers for the task. The volunteers were offered a small amount of money in exchange for their services, and they were required to be Malaysian citizens and able to commit to spending two separate days (one day for each forest) and several hours hiking through rugged forest terrain. The researchers stipulated that volunteers must be able to manage the physical demands of the task and be of the appropriate age and level of physical fitness. However, the audience response resulted in a limited number of volunteers. Twelve candidates were chosen from the public who have expressed a desire to help (four men and eight women), who were all Malaysians of diverse ethnicities (Malay, Chinese, and Indian). The participants ranged from 29 to 33 years old, indicating that they could handle exertion and spend multiple hours in the forest. The participants were distinguished by their different studies, which included: landscape architecture, architecture, human ecology, forestry, communication, computer science, law, education, and language. They were evenly divided among four distinct educational backgrounds, which included: design, environmental, technical, and social. Researchers used purposive sampling to select equal background groups in order to compare across backgrounds more accurately. It is a technique utilized extensively in research for the identification

and selection of information-rich cases in order to make more effective use of limited samples [47] (Table 3).

**Table 3.** Twelve participants of the public were recruited based on their background for data collection.

| Photographer No. | Gender | Age | Ethnicity | Education | Background |
|---|---|---|---|---|---|
| 1 | Female | 30 | Chinese | Landscape architecture | Design |
| 2 | Female | 29 | Malay | Landscape architecture | Design |
| 3 | Male | 29 | Chinese | Architecture | Design |
| 4 | Female | 30 | Indian | Ecology | Environmental |
| 5 | Female | 30 | Chinese | Forestry | Environmental |
| 6 | Female | 30 | Chinese | Forestry | Environmental |
| 7 | Male | 29 | Malay | Communication engineer | Technical |
| 8 | Male | 29 | Malay | Computer Science | Technical |
| 9 | Male | 33 | Malay | Computer Science | Technical |
| 10 | Female | 30 | Indian | Law | Social |
| 11 | Female | 30 | Chinese | Education | Social |
| 12 | Female | 30 | Malay | Language | Social |

We used participant-generated image (PGI) methods to capture the scenes participants liked along the trail using smartphone cameras [34]. Capturing images based on the preferences of the participants is an effective way to validate the variables extracted from prior studies. One of the most valuable aspects of PGI is the participants' visual attention and aesthetic preference rather than the use of text [48,49]. We asked participants to photograph views they thought were beautiful while adhering to these constraints: the pictures had to be devoid of people, the camera had to be at eye level, and the pictures had to be of the scenery rather than of a specific object.

### 2.3. Data Classification

This study depends on a converging approach to identifying suitable variables for visual aesthetic quality based on public and expert opinion [3,34]. Researchers and visual aesthetic assessment experts in the landscape architecture department attempted to identify the most important variables for assessing the visual aesthetic quality of permanent forest reserves in urban areas in the Klang Valley based on the collected PGI data by participant preferences in two stages (Figure 3). In the first stage, the researchers defined visual aesthetic variables based on theories of visual aesthetics extracted through the literature review and classified the photos by the potential variables to determine if the forest scenes in urban areas matched all the extracted variables.

In the second stage, experts identified the essential variables in three steps: validation of researcher classification, sub-classification, and reclassification. They verified and assessed the validity of the researcher's classification, with or without an agreement. Then, they reclassified the photos on which they could not agree in the previous step, utilizing the same seven variables extracted by the researcher. The contrasting opinion in the classification of the photos is reasonable, for some photos contained multiple variables incorporated into a single scene; however, the experts should choose the single most dominant visual aesthetic variable. Finally, they worked on the new classification and identified suitable visual aesthetic variables based on the content of permanent forest reserves in urban areas in the Klang Valley.

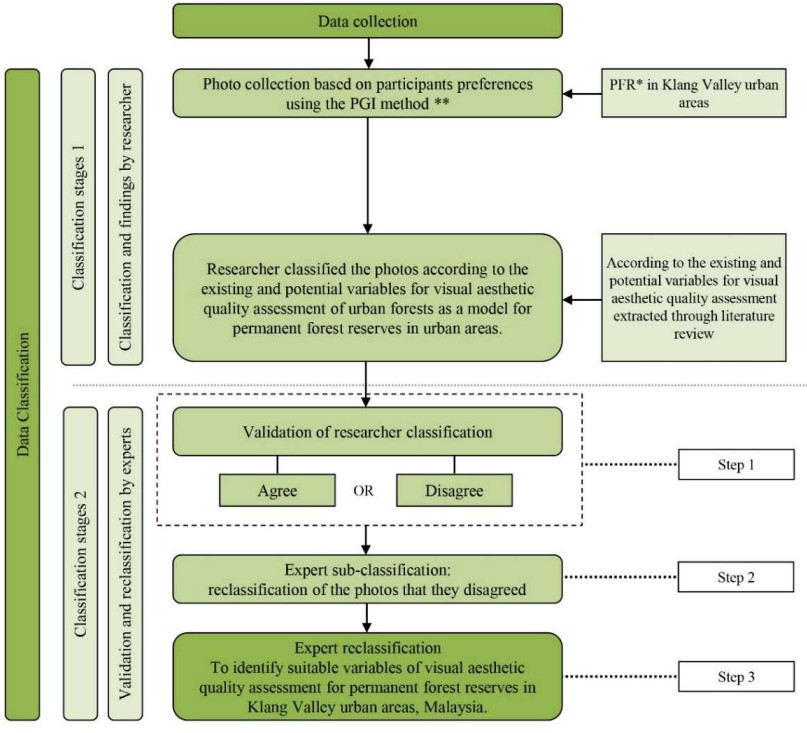

**Figure 3.** Framework for identifying visual aesthetic quality variables of permanent forest reserves in Klang Valley urban area, Malaysia.

### 2.4. Data Analysis

This study utilized the "Orange Data Mining" software to analyze the data because, unlike other statistical tools such as SPSS, it effectively analyzes small data [50]. Orange Data Mining is open-source artificial intelligence (AI) software that is freely accessible and useful for exploratory data analytics and visualization, providing a platform for various experiment selections. It was developed by the University of Ljubljana in Slovenia and designed to be user-friendly. According to previous research [50,51], orange is highly effective when dealing with the concepts of innovation, reliability, and quality.

In this software, feature statistics (descriptive statistics) and box plots were employed for data analysis. Furthermore, a heat map was generated based on the data to determine the correlation between the four distinct participant backgrounds and to determine the most correlated variables with each background.

## 3. Results

### 3.1. Researcher Classification and Findings

In PGI research, the number of images taken by participants is more important than the number of participants. In the bulk of previous studies, 100–500 photos have been shown to be the most common [48]. A specific number of photos are requested from participants in some studies, but we did not direct participants to take a certain number of photos, as this would restrict them to a certain range and prevent them from taking all of their preferred photos. We collected a total of 750 photos from 12 individuals, an average of 62 photos per participant for both study regions. Subsequently, the researcher classified the photos using seven visual aesthetic quality variables derived from several theories: coherence, complexity, legibility, mystery, openness, uniqueness, and cleanliness (Table 4).

**Table 4.** Researcher classification of visual aesthetic variables in photos of natural forests in Klang Valley urban area, Malaysia.

| Photographer No. | Coherence | Complexity | Legibility | Mystery | Openness | Uniqueness | Cleanliness | Total |
|---|---|---|---|---|---|---|---|---|
| 1 | 10 | 11 | 10 | 09 | 14 | 01 | 02 | 57 |
| 2 | 09 | 11 | 09 | 15 | 02 | 15 | 03 | 64 |
| 3 | 09 | 07 | 05 | 02 | 23 | 10 | 04 | 60 |
| 4 | 24 | 18 | 03 | 12 | 01 | - | 01 | 59 |
| 5 | 14 | 23 | 08 | 16 | - | 01 | 05 | 67 |
| 6 | 30 | 21 | 02 | 08 | - | 07 | - | 68 |
| 7 | 14 | 05 | 15 | 09 | 04 | 07 | 02 | 56 |
| 8 | 13 | 08 | 17 | 20 | - | 04 | 04 | 66 |
| 9 | 14 | 17 | 19 | 19 | 01 | - | 04 | 74 |
| 10 | 15 | 11 | 11 | 18 | 03 | - | - | 58 |
| 11 | 18 | 08 | 21 | 09 | - | 01 | 02 | 59 |
| 12 | 04 | 03 | 20 | 21 | - | 10 | 04 | 62 |
| Total | 174 | 143 | 140 | 158 | 48 | 56 | 31 | 750 |
| Percentage * | 23.2% | 19.0% | 18.7% | 22.0% | 6.4% | 7.5% | 4.1% | 100% |
| | | | | 615, 82.9% ** | | | | |

* Percentage = (number/total number) × 100%. ** Information-processing theory variables.

The researcher's preliminary photo classifications revealed a dominance of visual aesthetic variables derived from information-processing theory at 615 photos (82.9%) as divided into coherence (174 photos, 23.2%), mystery (158 photos, 22.0%), complexity (143 photos, 19.0%), and legibility (140 photos, 18.7%). While the cleanliness variable was one of the least used variables for expressing the aesthetics of natural forests in urban areas in Malaysia (31 photos, 4.1%). The researcher's primary findings are suitable in determining if the variables collected from visual aesthetic theories are present in the forest scene. It is also a prerequisite in initiating an expert study to identify the crucial visual aesthetic quality variables of natural forests in urban areas in the Klang Valley.

*3.2. Expert Validation, Sub-Classification, and Reclassification*

3.2.1. Validation of Researcher Classification

The experts agreed with the researchers on 514 photos (68.5%) and disagreed on 236 photos (31.5%) (Table 5). There was agreement in classification on more than two-thirds of the total photos, which demonstrates the validity of the researcher's preliminary findings. The contrasting opinions among the experts and researchers regarding the remaining photos are understandable because some of the images contained multiple variables incorporated into a single scene. This complexity posed a challenge for the experts, who had to identify and choose a single dominant visual aesthetic variable to accurately capture the scene's essence. Despite this difficulty, their expertise aimed to guarantee the most representative and meaningful assessment of the visual aesthetic quality of urban forest reserves.

**Table 5.** Expert agreement and disagreement of visual aesthetic variable classification of the researcher.

| Photographer No. | Coherence | | Complexity | | Legibility | | Mystery | | Openness | | Uniqueness | | Cleanliness | | Total |
|---|---|---|---|---|---|---|---|---|---|---|---|---|---|---|---|
| | A | D | A | D | A | D | A | D | A | D | A | D | A | D | |
| 1 | 05 | 05 | 03 | 08 | 10 | - | 08 | 01 | 12 | 02 | 01 | - | - | 02 | 57 |
| 2 | 01 | 08 | 03 | 08 | 07 | 02 | 14 | 01 | 01 | 01 | 08 | 07 | - | 03 | 64 |
| 3 | 04 | 05 | 04 | 03 | 03 | 02 | 02 | - | 16 | 07 | 04 | 06 | - | 04 | 60 |
| 4 | 16 | 08 | 14 | 04 | 02 | 01 | 11 | 01 | - | 01 | - | - | - | 01 | 59 |

**Table 5.** *Cont.*

| Photographer No. | Coherence | | Complexity | | Legibility | | Mystery | | Openness | | Uniqueness | | Cleanliness | | Total |
|---|---|---|---|---|---|---|---|---|---|---|---|---|---|---|---|
| | **A** | **D** | **A** | **D** | **A** | **D** | **A** | **D** | **A** | **D** | **A** | **D** | **A** | **D** | |
| 5 | 10 | 04 | 14 | 09 | 07 | 01 | 13 | 03 | - | - | 01 | - | - | 05 | 67 |
| 6 | 27 | 03 | 16 | 05 | - | 02 | 07 | 01 | - | - | 06 | 01 | - | - | 68 |
| 7 | 05 | 09 | 04 | 01 | 10 | 05 | 07 | 02 | 04 | - | 05 | 02 | - | 02 | 56 |
| 8 | 03 | 10 | 04 | 04 | 16 | 01 | 20 | - | - | - | 04 | - | - | 04 | 66 |
| 9 | 12 | 02 | 12 | 05 | 15 | 04 | 17 | 02 | 01 | - | - | - | - | 04 | 74 |
| 10 | 13 | 02 | 06 | 05 | 09 | 02 | 14 | 04 | 03 | - | - | - | - | - | 58 |
| 11 | 08 | 10 | 01 | 07 | 13 | 08 | 07 | 02 | - | - | 01 | - | - | 02 | 59 |
| 12 | 04 | - | 02 | 01 | 14 | 06 | 20 | 01 | - | - | 10 | - | - | 04 | 62 |
| Total | 108 | 66 | 83 | 60 | 106 | 34 | 140 | 18 | 37 | 11 | 40 | 16 | 00 | 31 | 750 |

A = agree, D = disagree.

### 3.2.2. Sub-Classification

The researcher and expert opinions agreed on the dominance of four variables, which were the information-processing theory variables in 643 photos (84.5%) as divided into mystery (237 photos, 31.6%), legibility (165 photos, 22.0%), coherence (135 photos, 18.0), and complexity (97 photos, 12.9%) (Table 6). Mystery was the dominant variable between them, suggesting that the participants favored the promised exploration that could be achieved if the viewer relocated to a different area without feeling scared or in danger. On the other hand, the experts identified a few photos that express the variable of openness (44 photos, 5.8%) and uniqueness (72 photos, 9.7%).

**Table 6.** Expert sub-classification of visual aesthetic variables in photos of permanent forest reserves in Klang Valley urban area, Malaysia.

| Photographer No. | Coherence | Complexity | Legibility | Mystery | Openness | Uniqueness | Cleanliness | Total |
|---|---|---|---|---|---|---|---|---|
| 1 | 05 | 03 | 19 | 16 | 12 | 02 | - | 57 |
| 2 | 03 | 03 | 20 | 25 | 04 | 09 | - | 64 |
| 3 | 12 | 10 | 04 | 13 | 16 | 05 | - | 60 |
| 4 | 19 | 17 | 04 | 14 | - | 05 | - | 59 |
| 5 | 11 | 14 | 16 | 17 | - | 09 | - | 67 |
| 6 | 31 | 17 | - | 10 | - | 10 | - | 68 |
| 7 | 05 | 04 | 12 | 20 | 08 | 07 | - | 56 |
| 8 | 03 | 04 | 23 | 32 | - | 04 | - | 66 |
| 9 | 16 | 14 | 18 | 23 | 01 | 02 | - | 74 |
| 10 | 14 | 07 | 16 | 15 | 03 | 03 | - | 58 |
| 11 | 11 | 01 | 19 | 22 | - | 06 | - | 59 |
| 12 | 05 | 03 | 14 | 30 | - | 10 | - | 62 |
| Total | 135 | 97 | 165 | 237 | 44 | 72 | 00 | 750 |
| Percentage * | 18.0% | 12.9% | 22.0% | 31.6% | 5.8% | 9.7% | 0% | 100% |
| | | | 634, 84.5% ** | | | | | |

\* Percentage = (number/total number) × 100%. \*\* Information-processing theory variables.

Cleanliness was not a dominant variable in the photos taken by the participants since the experts preferred the scenes due to the dominance of other variables. Although cleanliness was present in the majority of the photos, the experts did not see this as a primary reason for visual aesthetic preference. The variables affecting the participants were those based on the composition and a sense of nature that are more aesthetically pleasing when no significant human intervention is present. Therefore, the experts decided to exclude the cleanliness variable from the list of visual aesthetic variables for the permanent forest reserves in urban areas in Klang Valley.

At the conclusion of sub-classification phase, experts agreed on six primary assessment variables: coherence, complexity, legibility, mystery, openness, and uniqueness.

### 3.2.3. Reclassification

Experts identified 14 variables derived from the six major variables based on the content of photos for permanent forest reserves in urban areas in Klang Valley: coherence, complexity, legibility with natural path, legibility with man-made path, mystery with natural path, mystery with man-made path, openness with trees direct view, openness with trees frame view, openness with city view, openness with water view, uniqueness with natural elements, uniqueness with man-made elements, uniqueness water with natural elements, and uniqueness water with man-made elements (Table 7).

**Table 7.** Expert reclassification of visual aesthetic variables in photos of permanent forest reserves in Klang Valley urban area, Malaysia.

| Photographer No. | Coherence | Coherence | Legibility with Natural Path | Legibility with man-Made Path | Mystery with Natural Path | Mystery with Man-Made Path | Openness with Trees-Direct View | Openness with Trees-Frame View | Openness with City View | Openness with Water View | Uniqueness with Natural Elements | Uniqueness with Man-Made Elements | Uniqueness water with Natural Elements | Uniqueness Water with Man-Made Elements | Total |
|---|---|---|---|---|---|---|---|---|---|---|---|---|---|---|---|
| 1 | 05 | 03 | - | 19 | 06 | 10 | 07 | 05 | - | - | 01 | 01 | - | - | 57 |
| 2 | 03 | 03 | - | 20 | 17 | 08 | - | 01 | - | 03 | 01 | - | - | 08 | 64 |
| 3 | 12 | 10 | 01 | 03 | 07 | 06 | 02 | 01 | 13 | - | - | 05 | - | - | 60 |
| 4 | 19 | 17 | - | 04 | 12 | 02 | - | - | - | - | 05 | - | - | - | 59 |
| 5 | 11 | 14 | - | 16 | 14 | 03 | - | - | - | - | 09 | - | - | - | 67 |
| 6 | 31 | 17 | - | - | 09 | 01 | - | - | - | - | 04 | - | 06 | - | 68 |
| 7 | 05 | 04 | - | 12 | 07 | 13 | 03 | 03 | - | 02 | 02 | - | 02 | 03 | 56 |
| 8 | 03 | 04 | - | 23 | 23 | 09 | - | - | - | - | - | - | 04 | - | 66 |
| 9 | 16 | 14 | - | 18 | 14 | 08 | 01 | - | - | - | - | 01 | 01 | - | 74 |
| 10 | 14 | 07 | 01 | 15 | 08 | 07 | - | 03 | - | - | 01 | 02 | - | - | 58 |
| 11 | 11 | 01 | 05 | 14 | 09 | 13 | - | - | - | - | - | 06 | - | - | 59 |
| 12 | 05 | 03 | 02 | 12 | 12 | 18 | - | - | - | - | - | 01 | 09 | - | 62 |
| Total | 135 | 97 | 09 | 156 | 138 | 98 | 13 | 13 | 13 | 5 | 23 | 16 | 22 | 11 | 750 |
| Percentage * | 18.0 | 12.9 | 1.2 | 20.8 | 18.5 | 13.1 | 1.7 | 1.7 | 1.7 | 0.7 | 3.1 | 2.1 | 3.0 | 1.5 | 100% |

* Percentage = (number/total number) × 100%.

The expert opinions agreed on the dominance of five variables (624 photos, 83.3%), which were divided into legibility with man-made path (156 photos, 20.8%), mystery with natural path (138 photos, 18.5%), coherence (135 photos, 18.0%), mystery with man-made path (98 photos, 13.1%), and complexity (97 photos, 12.9%). On the other hand, the experts identified that the least present variable is openness with water view (5 photos, 0.7%).

Also, it should be noted that the most significant proportion of participants who chose the openness variables among participants (number 1, 2, and 3); specifically participant number 3, with an architectural background, was the only individual who climbed to the top of the forest to take photos classified as openness with the city view (13 photos, 1.7%).

At the conclusion of the reclassification phase, experts identified 14 suitable variables for the visual aesthetic quality assessment of permanent forest reserves in urban areas in Klang Valley, Malaysia (Table 10).

**Table 8.** A photo and description of each of the 14 variables used to define the visual aesthetic assessment of the permanent forest reserves within urban areas.

| No. | Variable | Description | Photo |
|---|---|---|---|
| 1 | Coherence | The scene's elements unite and cohere to produce an immediate understanding of the two-dimensional plane. |  |
| 2 | Complexity | The scene's elements diverse and varied non-chaotic to produce an immediate exploration of the two-dimensional plane. |  |
| 3 | Legibility with natural path | The scene promises an understanding of direction and accessibility in three-dimensional space with a natural path. |  |
| 4 | Legibility with man-made path | The scene promises an understanding of direction and accessibility in three-dimensional space with a man-made path. |  |

**Table 8.** *Cont.*

| No. | Variable | Description | Photo |
|---|---|---|---|
| 5 | Mystery with natural path | The scene promises exploration and motivation for the viewer to discover hidden elements in three-dimensional space with a natural path. |  |
| 6 | Mystery with man-made path | The scene promises exploration and motivation for the viewer to discover hidden elements in three-dimensional space with a man-made path. |  |
| 7 | Openness with trees-direct view | The scene has a vast field of vision, through which an observer can perceive the vastness of the surrounding area with trees-direct view. |  |
| 8 | Openness with trees-frame view | The scene has an open perspective line of vision with tree borders on both sides that resemble a frame. |  |
| 9 | Openness with city view | The scene has a panoramic view, from which an observer can feel the vastness of the surrounding area that includes city views from within the forest. |  |

**Table 8.** *Cont.*

| No. | Variable | Description | Photo |
|---|---|---|---|
| 10 | Openness with water view | The scene has a panoramic view, from which an observer can feel the vastness of the surrounding area that includes water view. |  |
| 11 | Uniqueness with natural elements | The scene's natural elements provide a distinct visual impression and are memorable. |  |
| 12 | Uniqueness with man-made elements | The scene's man-made elements provide a distinct visual impression and are memorable. |  |
| 13 | Uniqueness water with natural elements | The scene's water with natural elements provides a distinct visual impression and are memorable. |  |
| 14 | Uniqueness water with man-made elements | The scene's water with man-made elements provides a distinct visual impression and are memorable. |  |

### 3.3. Differences in Responses Amongst Participants' Various Backgrounds

When summing the number of photos taken by each group of three participants with the same background, a similarity emerged in the total number of photos for each background. It was distributed as follows: technical (196 photos, 26.1%), environmental (194 photos, 25.9%), design (181 photos, 24.1%), and social (179 photos, 23.9%), as shown in Table 9.

**Table 9.** Total photo statistics of the visual aesthetic variables based on the participants' backgrounds.

|  | Variables | Design | Environmental | Technical | Social |
|---|---|---|---|---|---|
| 1 | Coherence | 20 | 61 | 24 | 30 |
| 2 | Complexity | 16 | 48 | 22 | 11 |
| 3 | Legibility with natural path | 1 | 0 | 0 | 8 |
| 4 | Legibility with man-made path | 42 | 20 | 53 | 41 |
| 5 | Mystery with natural path | 30 | 35 | 44 | 29 |
| 6 | Mystery with man-made path | 24 | 6 | 30 | 38 |
| 7 | Openness with trees-direct view | 9 | 0 | 4 | 0 |
| 8 | Openness with trees-frame view | 7 | 0 | 3 | 3 |
| 9 | Openness with city view | 13 | 0 | 0 | 0 |
| 10 | Openness with water view | 3 | 0 | 2 | 0 |
| 11 | Uniqueness with natural elements | 2 | 18 | 2 | 1 |
| 12 | Uniqueness with man-made elements | 6 | 0 | 1 | 9 |
| 13 | Uniqueness water with natural elements | 0 | 6 | 7 | 9 |
| 14 | Uniqueness water with man-made elements | 8 | 0 | 3 | 0 |
|  | Total | 181 | 194 | 196 | 179 |
|  | Percentage * | 24.1% | 25.9% | 26.1% | 23.9% |

* Percentage = (number/total number) × 100%.

Similarly, the descriptive statistics of participants from different backgrounds revealed a convergence of the means, sequenced as follows: technical (M = 13.93), environmental (M = 13.86), design (M = 12.93), and social (M = 12.79). Furthermore, the findings demonstrate a high level of reliability, as indicated by the total R value of 0.85, indicating good internal consistency and reliability. In addition, the findings indicate that the "coherence" variable has the highest value within the environmental background, with a total of 61 photos. In contrast, within the technical, design, and social backgrounds, the "legibility with man-made path" variable yields the highest values, totaling 53, 42, and 41 photos, respectively, as shown in Table 10.

**Table 10.** Descriptive statistics for different participants' backgrounds.

| Background | Mean | Median | Min. | Max. | Reliability | Box Plot |
|---|---|---|---|---|---|---|
| Design | 12.93 | 8.50 | 0 | 42 — Legibility with man-made path | 0.83 | 12.93 ± 11.9; 4.50, 8.50, 18 |
| Environmental | 13.86 | 3.00 | 0 | 61 — Coherence | 0.95 | 13.86 ± 19.6; 0, 3, 19 |

**Table 10.** *Cont.*

| Background | Mean | Median | Min. | Max. | Reliability | Box Plot |
|---|---|---|---|---|---|---|
| Technical | 13.93 | 3.50 | 0 | 53 Legibility with man-made path | 0.78 | 13.93 ± 17.1 / 2 3.50 / 23 |
| Social | 12.79 | 8.50 | 0 | 41 Legibility with man-made path | 0.82 | 12.79 ± 14.5 / 0.50 8.50 20 |
| | | | | | R-value 0.85 | |

The results of the heat map correlation analysis for participants with diverse backgrounds reveal a strong correlation between design and technical backgrounds. In addition, there is a strong correlation between the design and technical backgrounds and the social backgrounds. In contrast, the environmental background demonstrates a weak correlation with the other backgrounds.

In addition, the analysis of the heat map reveals correlations between variables and different backgrounds. In particular, "legibility with man-made path," "mystery with natural path," and "mystery with man-made path" have a significant impact on the design, technical, and social backgrounds. Conversely, the variables "coherence" and "complexity" exhibit the most prominent impact on the environmental background (Figure 4).

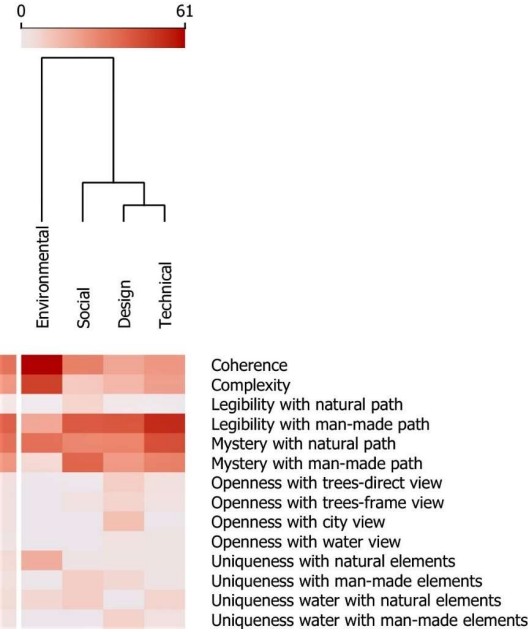

**Figure 4.** Heat map analyze of variables with different participants' backgrounds.

## 4. Discussion

The seven variables of perception of visual aesthetic quality were extracted from visual aesthetic theories to identify suitable variables of permanent forest reserves in urban areas in Malaysia. The seven primary variables of visual aesthetics via aesthetic theories are coherence, complexity, legibility, mystery, openness, uniqueness, and cleanliness. The information-processing theory was the most definitive theory for visual aesthetic quality assessment of permanent forest reserves in urban areas due to the dominance of four variables: coherence, complexity, legibility, and mystery. The reason for a large number of visual aesthetic variable photos is that the visual characters of natural forests in urban areas in Malaysia stem from two basic needs: a clear understanding of the information that reaches the viewer and, thus, a sense of comfort and aesthetic understanding, and the motivation of exploration that the scene promises if the viewer moves to another location in the forest. Mystery is the most present variable for visual aesthetic quality assessment of permanent forest reserves in urban areas, indicating that the viewer is interested in discovering hidden elements. This is consistent with Kaplan's [31] opinion that humans evolved in a way that makes us attracted to missing information or the unknown. Psychologically, the viewer's preference for mystery might be related to curiosity in discovering what is concealed or the human desire to discover without feeling scared or in danger [52]. In addition, cleanliness was not a dominating variable in the photos taken by the participants. This could be due to the Malaysian perceiving natural forests in urban areas as more beautiful when there is no evident human intervention, unwillingness to convert forests into artificial green areas, or the desire of Malaysian citizens to see fallen leaves and trees as natural parts of aesthetics for natural forests in urban areas, This is consistent with Kirillova [41]. Also, human trash was absent from all photos due to the care taken by individuals responsible for preserving natural forests in urban areas and the Malaysian culture of not littering in natural forests in urban areas. Therefore, it was decided to remove the cleanliness variable from the list of visual aesthetic variables for permanent forest reserves in urban areas in Klang Valley.

In conjunction with the identification of 14 suitable variables for assessing the visual aesthetic quality of permanent forest reserves in urban areas, the study found the educational background of the participants had a clear impact on visual aesthetic preferences. Those with design, technical, and social backgrounds correspond to preferred nature and man-made scenes with 3D elements "legibility and mystery variables with nature and man-made scenes of the 3D space described by Kaplan [31]". All participants with these backgrounds expressed the same opinions and preferences through their preference for what may be perceived when a person moves to a new location, whether the information is clear or hidden. These participants preferred permanent forest reserves in urban area sceneries with 3D trails, given their comparable educational backgrounds. Conversely, environmental backgrounds preferred coherence and complexity. This shows that participants prefer tangible, understandable information, regardless of its complexity. These participants preferred 2D natural environments devoid of trails. They confirmed again their preference for uniqueness with natural elements more than those from other backgrounds due to their experience with and education in relation to ecosystems.

Participants with a design background preferred the openness variable more than the rest of the participants. This may be due to the fact that the concept of openness provided a wide panoramic view, one of the terms frequently used in the design process of forests. People enjoy open spaces and unimpeded horizons, and the wider the range of view, the higher ranking their aesthetic preference; these results are consistent with Mundher [34] and Gao [53]. In contrast, the environmental background did not prefer openness because their studies focused on plant density and diversity.

Participant preferences for uniqueness varied as a result of their diverse cultural and educational backgrounds. Different elements created distinct visual impressions in their memories and made the forest scenes unique and unforgettable for each participant. Visual aesthetic preference of uniqueness depended on viewer sentiment, emotion, and back-

ground (push factors), while interactions between urban residents and permanent forest reserves in urban areas influenced their perceptions, preferences, and selection of forests to represent the community, cultural, and aesthetic values (pull factors) [23]. In a direct sense, the emotions and expertise allowed them to view a selection of natural components as outstanding, indicating that educational and emotional background influenced participants' aesthetic preferences; these results are consistent with Nazemi [15] and Mundher [54]. These results support prior findings that personal beliefs and values influence aesthetic preferences [55–57].

## 5. Limitations and Future Studies

This study produced novel and pertinent findings, although it has some limitations. We concentrated on assessing natural forests in urban areas classified as PFRs. A logical next step would be to apply these variables to natural forests with non-urban areas and determine if there is a difference between the variables for urban and natural forests. This study was conducted with 12 participants, split evenly across design, environmental, technical, and social backgrounds. No participants with a medical background were included due to a lack of volunteers. We anticipate differences in preference for visual aesthetics based on their different educational backgrounds, which is worth investigating. Also, the response of volunteers in this study was limited to individuals between the ages of 29–33. Therefore, we recommend conducting expanded quantitative studies in the future that include participants from different age groups, allowing for a comparison of the results with our study. Also, it is worth noting that due to the vast amount of trails in permanent forest reserves in urban areas, no specific trail was selected for the participants and the participants chose the path they wished to take. As a result, some paths were disregarded by some of the participants.

## 6. Conclusions

This study identified 14 suitable variables for assessing the aesthetic quality of permanent forest reserves in urban areas in Klang Valley, Malaysia. This study depends on a converging approach to identifying suitable variables for visual aesthetic quality based on expert and public opinion. Although the goal of this study was to identify suitable variables to assess the visual aesthetic quality of PFRs in urban areas, we found that the educational and emotional background of the participants directly influenced their aesthetic preferences. The educational background of the participants had a clear impact on visual aesthetic preferences, including that those with design, technical, and social backgrounds expressed the same opinions and preferences through their preference for what may be perceived when a person moves to a new location, whether the information is clear or hidden. These participants preferred permanent forest reserves in urban area sceneries with three-dimensional trails, given their comparable educational backgrounds. In contrast, those with an environmental background preferred tangible, understandable information, regardless of its complexity. These participants preferred two-dimensional natural environments devoid of trails due to their experience with and education in relation to ecosystems.

This study reveals the novelty of a new variable "openness with city view", which, to the best of our knowledge, is not found in other studies of the visual aesthetic assessment of PFRs in urban areas. Openness with city view evaluates the visual aesthetic quality of the view of the city from within the forest and holds the potential to help understand the visual aesthetics of PFRs in urban areas and stop the urbanization of forested areas. Future research should focus on an aesthetically suitable visual distance that must be sustained to view a city.

Even though we identified 14 major variables to assess the visual aesthetics of PFRs in urban areas in Klang Valley, we still need to determine the relative weights of these variables as we believe that these variables are not of equal importance. Therefore, we invite researchers to develop a framework based on expert opinion to determine the weight of

each variable and the degree to which each variable dominates the scene's characterization. In addition, we advocate comparing the preference results of experts and the public to see whether there is a distinction in visual aesthetic assessment. This study highlights suitable variables for visual aesthetic assessment in PFRs in urban areas, which will help researchers, designers, forest managers, and decision-makers manage and safeguard the visual aesthetic value of permanent forest reserves in urban areas in Malaysia.

**Author Contributions:** Conceptualization, R.M. and S.A.B.; methodology, R.M.; validation, S.A.B. and S.M.; formal analysis, R.M., S.A.B. and S.M.; investigation, R.M., H.G. and A.A.-S.; resources, R.M.; data curation, R.M., H.G. and A.A.-S.; writing—original draft preparation, R.M.; writing—review and editing, S.A.B. and A.A.; visualization, R.M.; supervision, M.J.M.Y. and A.A.; project administration, M.J.M.Y. All authors have read and agreed to the published version of the manuscript.

**Funding:** This research received no external funding.

**Data Availability Statement:** No data available.

**Conflicts of Interest:** The authors declare no conflict of interest.

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
