# Peer review of "Identifying Suitable Variables for Visual Aesthetic Quality Assessment of Permanent Forest Reserves in the Klang Valley Urban Area, Malaysia"

_urbansci, doi:10.3390/urbansci7030092_

Round 1

Reviewer 1 Report

This is a very interesting study.But there are still some problems to the manuscript in some details, we need to further optimize and improve.

1.Line 74-76: The reasons for studying visual aesthetics are not sufficient and further refinement is needed.

2.Line 258: Why are the ages of volunteers concentrated between 29 and 33? More information on volunteer recruitment needs to be made public.

3.In terms of methodology, evaluating the quality of visual aesthetics relies on expert experience, will this limit the accuracy and rationality of the evaluation? Whether to consider using quantitative evaluation methods to reduce reliance on experts.

Reviewer 2 Report

The manuscript describes a research study aimed at identifying suitable variables for assessing the visual aesthetic quality of permanent forest reserves within urban areas in Malaysia. After review, I have some comments as follows.

·        In section 1.1 it would be helpful to describe some context of natural forests in urban areas in Malaysia and the linkages to reviewed studies.

·        Table 1, the reference "Information processing theory [29,30]" should be included for variables 2, 3, and 4 as well.

·        Figure 1's resolution appears to be too low, and it should highlight the two selected study areas.

·        The study exhibits selection bias issues. When participants are predominantly volunteers, the research is likely to have skewed results.  The use of purposive sampling to create equal background groups also introduces bias. Moreover, the total number of participants, number of groups, and distribution of participants within each group lack justification.

·        To enhance comprehensiveness, Figures 3 and 5 are unnecessary as the information presented in these figures can be clearly understood from the text.

·        The data classification section should be rewritten to enhance clarity and comprehensiveness. The sentence "The researcher defined visual aesthetic variables based on theories and gathered photos based on participant preferences" is confusing due to the repeated use of 'based on.' Additionally, the roles of other researchers involved need clarification, as the claim of only one researcher conducting the first stage appears inconsistent. Furthermore, "on the previous stage" in line 293 should be either "from the previous stage" (when the researcher performed the classification) or "in the previous step" (when experts disagreed on the photo classification).

·        The strategy for dealing with photos containing multiple variables should be clearly described in the methods section rather than later sections.

·        Authors should consider rewriting the result and discussion sections. As of now, the results section has been divided into 3 subsections for 3 steps of classification, however, each employs different approaches and logic. Subsection 3.1 explained the pattern of variable preferences among all participants, irrespective of their background. In contrast, subsection 3.2.2 specifically focused on participants with different backgrounds. Section 4 (discussion) just merely provides a summary of discussions already derived from the results section.  For example, the openness variable was dominant for people who have design background in both researcher's and expert’s classification but was not mentioned in subsection 3.1, only described in subsection 3.2.3 and then repeated in section 4.  

 I think authors should have a results section that simply presents the findings in an academic and unbiased manner, avoiding any attempt at analysing, explaining, or interpreting them. Subsequently, provide focused discussions in Section 4 that analyzes, compares, and elucidates the results obtained from the researcher's classification, sub-classification, and the expert's reclassification. This well-structured approach serves to eliminate redundancy, enhance logical coherence, and improve the overall comprehensiveness of the manuscript.

 ·        The use of bold text in most tables is not consistent and explained. Table 3 also does not need the background column if it does not serve the arguments in subsection 3.1.

Overall, I believe the study has a sound research approach, and the experiment has been conducted properly. However, the manuscript still struggles to effectively present the ideas and results.

Round 2

Reviewer 1 Report

My comments had received a good response, so I think it's time to accept this manuscript.

Author Response

We appreciate your comprehensive review of our manuscript and are grateful for your valuable comments.

Reviewer 2 Report

The manuscript has undergone significant revisions and improvements. I just have a few minor suggestions for modifications, outlined below:

- In Figure 1, it is advisable to minimize the use of different text fonts and sizes. Additionally, consider enhancing the clarity of the highlighting for the two studied sites.

- Please consider omitting the following statement from line 413: "likely due to his propensity for integrating the forest with buildings more than the other participants." Its removal will help avoid the appearance of interpretation in the results section.

- I recommend a comprehensive rewrite of Section 3.3. Although the authors employed various statistical analyses, they did not elaborate on the most significant findings. For instance, the heat map only appears to offer an additional interpretation of Table 9. Also, it might be beneficial to merge Table 10 with Table 9 or remove one of them, to enhance comprehensiveness.

In conclusion, I support the publication of the manuscript. The aforementioned points are offered as suggestions, and the authors are free to decide whether to incorporate them or not.
